# Rhododendrin-Induced RNF146 Expression via Estrogen Receptor β Activation is Cytoprotective Against 6-OHDA-Induced Oxidative Stress

**DOI:** 10.3390/ijms20071772

**Published:** 2019-04-10

**Authors:** Hyojung Kim, Jisoo Park, HyunHee Leem, MyoungLae Cho, Jin-Ha Yoon, Han-Joo Maeng, Yunjong Lee

**Affiliations:** 1Division of Pharmacology, Department of Molecular Cell Biology, Sungkyunkwan University School of Medicine, Samsung Biomedical Research Institute, Suwon, Gyeonggi-do 440-746, Korea; hjung93@skku.edu (H.K.); zysu0728@skku.edu (J.P.); 2National Development Institute of Korean Medicine, Gyeongsan 38540, Korea; npb0391@nikom.or.kr (H.L.); meanglae@nikom.or.kr (M.C.); 3College of Pharmacy, Gachon University, Incheon 21936, Korea; jinha89@daum.net (J.-H.Y.); hjmaeng@gachon.ac.kr (H.-J.M.)

**Keywords:** rhododendrin, RNF146, estrogen receptor, poly (ADP-ribose), dopaminergic neurodegeneration, Parkinson’s disease

## Abstract

Ring finger protein 146 (RNF146) is an E3 ubiquitin ligase whose activity prevents poly (ADP-ribose) polymerase 1 (PARP1)-dependent neurodegeneration in Parkinson’s disease (PD). Previously, we reported that rhododendrin is a chemical inducer that increases RNF146 expression. However, the molecular mechanism of rhododendrin-induced RNF146 expression is largely unknown and its translational application for the treatment of Parkinson’s disease remains unexplored. Here we found that rhododendrin increased RNF146 expression via estrogen receptor β (ERβ) activation. Rhododendrin stimulated ERβ nuclear translocation and binding to the RNF146 promoter, thereby enhancing its transcription. Rhododendrin is cytoprotective against 6-hydroxydopamine (6-OHDA)-induced cell death, which is largely dependent on ERβ activity and RNF146 expression. Finally, we demonstrated that rhododendrin treatment resulted in RNF146 expression in dopaminergic neurons in mice. Moreover, dopaminergic neuron viability was markedly enhanced by pretreatment with rhododendrin in 6-OHDA-induced mouse models for PD. Our findings indicate that estrogen receptor activation plays a neuroprotective role and that rhododendrin could be a potential therapeutic agent in preventing PARP1-dependent dopaminergic cell loss in PD.

## 1. Introduction

Parkinson’s disease (PD) is characterized by the irreversible and progressive loss of dopaminergic neurons in the substantia nigra pars compacta [1,2]. This demise of dopamine producing neurons is responsible for the cardinal motor symptoms of affected PD patients [1,2]. Several cell death mechanisms have been identified in the execution of dopaminergic neuron death in PD, including apoptosis, necrosis, and parthanatos, which is mediated by poly (ADP-ribose) polymerase 1 (PARP1) overactivation [3,4,5]. Parthanatos contributes to age dependent loss of dopaminergic neurons in PD mouse models and PARP1 activation has been observed in several neurodegenerative brain disorders and neurological insults in humans [3,6]. Because there are no treatments to halt or reverse degenerative process of dopaminergic neurons in PD, targeting molecular regulation of PARP1 overactivation and parthanatos has been considered a promising therapeutic strategy for PD treatment.

Ring finger protein 146 (RNF146) is an E3 ubiquitin ligase that specifically recognizes poly (ADP-ribose) (PAR)-conjugated protein substrates and targets them for proteasomal degradation [7]. In this regard, overstimulated PARP1, which is heavily self-PARylated, is a major target substrate for RNF146 [7,8]. RNF146 recognizes self-PAR conjugated PARP1 and can thus obstruct PARP1-mediated cell death pathways. Indeed, RNF146 expression in mouse brains has been shown to be neuroprotective in several disease models of PARP1 activation and neuronal loss [8,9]. Recently, we have discovered several RNF146 inducing small compounds, including liquiritigenin, rhododendrin, piperlonguminine, and chlorogenic acids. Liquiritigenin’s neuroprotective effects have been shown to be dependent on RNF146 expression and estrogen receptor β (ERβ) activation [9]. However, other RNF146 inducing compounds have not been characterized in depth.

Rhododendrin is an arylbutanoid glycoside, an analgesic and anti-inflammatory component from the leaves of *Rhododendron aureum* [10]. Rhododendrin was shown to exert its anti-inflammatory actions through inhibiting toll-like receptor 7 and NF-kB activation in animal models of skin inflammation [11,12]. Moreover, it was reported that rhododendrin has the ability to scavenge intracellular reactive oxygen species [10]. Although rhododendrin’s anti-inflammatory function could be beneficial in combatting the pathogenesis of neurodegenerative diseases, it has yet to be investigated whether rhododendrin could be a potential therapeutic agent for PD. 

Here, we show that treatment of rhododendrin, an arylbutanoid glycoside from *Rhododendron aureum,* is neuroprotective in PD mouse model. RNF146 expression induced by rhododendrin is mediated by estrogen receptor (ER) activation. Rhododendrin induction of RNF146 inhibits PARP1 activation and prevents cell death in in vitro PD models, suggesting a therapeutic potential for ERβ activating and RNF146-inducing compounds in PD.

## 2. Results

### 2.1. Rhododendrin-Induced RNF146 Expression Contributes to Cytoprotection Against Oxidative Stress

Rhododendrin was previously identified as a compound that has the ability to induce RNF146 expression [9]. To confirm this, we determined modulation of RNF146 promoter activity induced by rhododendrin (10 μM) using a RNF146-luciferase construct. SH-SY5Y cells were transiently transfected with pGL3-RNF146 Luc (2 kb RNF146 promoter) and pRL-Tk for transfection normalization. Rhododendrin treatment for 37 h increased RNF146 promoter activity (Figure 1A). Consistent with this observation, RNF146 messenger levels were also elevated in response to rhododendrin treatment (Figure 1B). There was a dose dependent enhancement of RNF146 mRNA expression, with an approximately five-fold induction by 10-µM rhododendrin treatment as compared to DMSO control group (Figure 1B). Next, we measured RNF146 protein expression via Western blot. Interestingly, 10-µM rhododendrin only led to a two-fold increase in RNF146 protein expression. 0.1 and 1-µM rhododendrin treatment for 37 h was not sufficient to elevate RNF146 protein expression (Figure 1C,D). However, extended treatment of rhododendrin (1, 5, 10 µM, 60 h) to SH-SY5Y cells resulted in dose-dependent increase of RNF146 protein expression (Appendix A), indicating delayed effect on protein translation relative to mRNA transcription by rhododendrin.

RNF146 expression is known to be cytoprotective via inhibition of PARP1 overactivity. Thus, we examined whether rhododendrin-induced RNF146 expression could prevent oxidative stress-induced cell toxicity. Treatment with 6-hydroxydopamine (6-OHDA; 70 µM, 16 h) of SH-SY5Y cells led to robust cell toxicity, with approximately 50% cell death occurring, as determined by a trypan blue exclusion assay. This 6-OHDA-induced cell death was largely prevented by pretreatment with rhododendrin (10 µM, 37 h) (Figure 2A). Supporting the role of PARP1 activation in 6-OHDA-induced cell death, there was a marked increase of poly (ADP-ribose) (PAR) conjugated proteins in total lysates from SH-SY5Y cells treated with 6-OHDA. This increase of PAR modified proteins was blocked by rhododendrin treatment, which correlated with RNF146 induction (Figure 2B–D). To determine whether rhododendrin-aided cell protection was mediated by RNF146 expression, endogenous RNF146 knockdown was achieved by shRNA transfection of SH-SY5Y cells. Trypan blue exclusion assays revealed that rhododendrin-mediated cytoprotection in 6-OHDA-challenged SH-SY5Y cells was abolished by RNF146 knockdown (Figure 2E), demonstrating that RNF146 induction is required for rhododendrin’s cytoprotective effect. Western blots confirmed robust PARP1 activation by 6-OHDA, which was prevented by rhododendrin (Figure 2F). shRNA against RNF146 efficiently knocked down endogenous RNF146 and prevented its expression by rhododendrin treatment (Figure 2F,G). When RNF146 expression was reduced with shRNA, PAR activation was maintained, even in the presence of rhododendrin (Figure 2F,H).

### 2.2. RNF146 Expression and Cytoprotection by Rhododendrin Requires ERβ Activation

We previously reported that ERβ activation contributes to liquiritigenin-induced RNF146 expression. To determine whether rhododendrin increases RNF146 expression via similar molecular mechanisms, we examined RNF146 mRNA levels in the presence of pharmacological inhibition of ERβ. ER inhibition by tamoxifen blocked RNF146 transcription induced by rhododendrin (Figure 3A), indicating that rhododendrin increased RNF146 expression via ERβ activation. Consistent with these results, tamoxifen-mediated ER inhibition abolished RNF146 protein induction by rhododendrin (Figure 3B,C). We further monitored the role of ERβ in RNF146 expression by using CRISPR-cas9 deletion of ERβ. Efficient knockdown of ERβ by CRISPR-cas9 transfection prevented rhododendrin-induced elevation of RNF146 mRNA and protein (Figure 3D–F).

Since we confirmed the important role of ERβ in mediating rhododendrin’s effect on RNF146 expression, we determined whether ERβ is indeed activated in response to rhododendrin treatment. ERβ activation was indirectly determined by monitoring nuclear translocation of ERβ using immunofluorescence. Rhododendrin drove nuclear translocation of ERβ in SH-SY5Y cells, indicating activation of ERβ (Figure 3G,H). We next investigated whether nuclear ERβ binds to the RNF146 promoter by using chromatin immunoprecipitation (ChIP) assays. Following treatment with rhododendrin, SH-SY5Y cells were subjected to anti-ERβ ChIP. Occupancy of the RNF146 promoter by ERβ was determined by PCR using specific primer sets for ChIPed DNA. The results showed an increase of ERβ binding to RNF146 promoter regions when SH-SY5Y cells were treated with rhododendrin (Figure 3I).

Given the significant role of ERβ activation after rhododendrin treatment, cell viability was again assessed under conditions of ERβ knockdown. CRISPR-cas9-mediated knockdown of ERβ resulted in failure of rhododendrin-induced prevention of 6-OHDA induced cytotoxicity (Figure 4A). Consistent with this cell viability assessment, ERβ deletion by CRISPR-cas9 blocked RNF146 induction by rhododendrin (Figure 4B–D). There were elevated levels of PAR modified proteins in total protein lysates when RNF146 expression was repressed by ERβ deletion compared to normal controls (Figure 4B,E).

### 2.3. Rhododendrin Prevents Dopaminergic Neuron Degeneration in a 6-OHDA PD Mouse Model

Rhododendrin is a phenolic glycoside with a cytoprotective function [10]. Polyphenolic extracts were shown to be neuroprotective in brains of Alzheimer’s mouse models. To determine whether rhododendrin treatment exerts a neuroprotective function in PD mouse models, we first sought to monitor expression of RNF146 in brains of mice treated with rhododendrin. In reference to the previous study [13], we chose 10 mg/kg rhododendrin intraperitoneal (i.p.) administration as a starting dose. Brain concentration of rhododendrin was measured by LC-MS/MS assay to determine whether rhododendrin penetrates into the brain tissue. When mouse brains were subjected to LC-MS/MS assay at 20 min after the second i.p. administration of rhododendrin (daily i.p. administration, 10 mg/kg), the rhododendrin concentration was estimated at 19.67 μg/g brain tissue (Appendix A). Intraperitoneal administration of rhododendrin for three days resulted in a 60% increase in RNF146 mRNA levels in the ventral midbrain (Figure 5A). There was also a three-fold elevation in RNF146 protein expression in the ventral midbrain induced by rhododendrin administration in vivo (Figure 5B,C). To see whether this increase of RNF146 expression indeed occurred in dopaminergic neurons, we performed immunofluorescent microscopy. Dopaminergic neurons were labelled with an anti-tyrosine hydroxylase (TH) antibody and endogenous mouse RNF146 was immunostained with anti-RNF146 specific antibodies. A robust increase in RNF146 immunoreactivity after rhododendrin treatment was observed in TH-positive dopaminergic neurons in the substantia nigra (Figure 5D,E).

Given the successful induction of RNF146 in dopaminergic neurons caused by rhododendrin treatment, we examined the neuroprotective effect of rhododendrin treatment in 6-OHDA-induced PD mouse models. Intrastriatal injection of the dopamine neurotoxin 6-OHDA led to more than 50% loss of TH-positive dopaminergic neurons. This loss was largely rescued by rhododendrin treatment (Figure 5F,G). Rhododendrin treatment in control mice had no detrimental effect on dopaminergic neuron viability (Figure 5F,G). Biochemical alterations were also determined using Western blots. Total protein lysates from the ventral midbrain of mice were prepared to examine the levels of PAR modified proteins and RNF146 expression. 6-OHDA-induced dopaminergic cell loss accompanied robust increase of PAR conjugated proteins (Figure 5H–J), indicating ongoing overactivation of PARP1. Rhododendrin treatment increased RNF146 expression and suppressed the PAR activation induced by intrastriatal injection of 6-OHDA (Figure 5H–J). Moreover, consistent with the western blot analysis, rhododendrin treatment increased RNF146 expression in TH-positive dopamine neurons in both 6-OHDA injection group and vehicle injection control group as determined by immunofluorescence (Appendix A) Taken together, our in vivo experiments demonstrated RNF146 expression and dopaminergic neuroprotective function by rhododendrin treatment in PD mouse models. 

## 3. Discussion

This is the first report to present evidence that rhododendrin could be a potential RNF146 inducer and, thus, a therapeutic compound for alleviating PD-associated dopaminergic neuron degeneration. PARP1 is a key mediator of a distinct cell death pathway in several neurodegenerative diseases [3,4,6,14]. Overactivation of PARP1 has been reported in animal models and postmortem brain tissues of neurodegenerative disorders, including PD [6,9]. Interestingly, inhibition of PARP1 overactivity was sufficient to block PD pathogenesis in PD mouse models of MPTP intoxication or AIMP2 overexpression [4,6]. Therefore, inhibiting the PARP1-related cell death pathway could lead to identification of potential therapeutic agents for treatment of PD. In this regard, chemical inhibitors against PARP have been developed and have provided substantial protective effects in ameliorating neurodegeneration in several mouse models of different brain disorders. However, PARP inhibitors block not only overstimulated PARP1 activity, but also basal activity of PARP1. Since basal activity of PARP1 is critically important in controlling gene transcription and maintaining genomic stability, prolonged exposure to PARP inhibitors could pose potential adverse effects when applied for neurodegenerative diseases that require long treatment plans. In this respect, targeting RNF146 might be a safer and more efficient strategy, as RNF146 selectively targets overactivated PARP1 without affecting inactive PARP1 [7]. Overstimulated and PAR conjugated PARP1 is specifically recognized by RNF146 and subsequently targeted for proteasomal degradation [7]. Rhododendrin treatment induced expression of RNF146 in cell culture and in vivo in mouse brains. Since we did not observe any overt toxicity with rhododendrin treatment, RNF146 induction and PARP1 inhibition by rhododendrin could serve as a safe therapeutic strategy to halt PARP1-dependent dopaminergic neuron demise in PD.

Rhododendrin has been studied for its therapeutic potential in animal models of skin inflammation [11,12]. Rhododendrin possesses antioxidant and anti-inflammatory activities [10]. Topical application of rhododendrin has been shown to repress inflammatory pathologies in psoriasis-like skin inflammation in mice. Additionally, skin hyperplasia, mononuclear cell recruitment, and proinflammatory mediators were ameliorated by administration of rhododendrin [12]. Our findings add additional functions to the previously characterized anti-inflammatory role of rhododendrin. The rhododendrin treatment led to RNF146 expression and suppressed dopaminergic cell loss in PD mouse models. However, it cannot be excluded that the anti-inflammatory function of rhododendrin partly contributed to its therapeutic effects in PD mouse models because neuroinflammation is also involved in the neurodegenerative process of PD. This possibility still needs further characterization. Interestingly, rhododendrin contains phenolic glycosides in its chemical structure. Several bioactive phenolic glycosides from *Markhamia stipulata* have shown cytoprotective activity that are connected to their antioxidant and chelating capacities, suggesting their potential treatment application for oxidative stress-related neurodegenerative diseases such as Alzheimer’s disease [15]. Rhododendrin’s cytoprotective effect is similar to these other phenolic glycosides. In this study, we showed that rhododendrin can penetrate into brain tissue, although there was large variation in the brain concentration of rhododendrin. This could be due to the brain sampling at non-steady state condition. It might be required to investigate thorough pharmacokinetic profiles of rhododendrin to determine tissue distribution and clearance of rhododendrin at steady state. Pharmacokinetic studies of rhododendrin will be imperative to further determine optimal and safe doses of rhododendrin and its potential application to several neurodegenerative disease animal models. 

RNF146 expression is regulated by preconditioning with a sublethal dose and brief duration of oxidative stress or DNA damage [8]. Previously, we reported that ER activation is involved in RNF146 transcription, even without the preconditioning paradigm [9]. Since RNF146 expression is suppressed and PARP1 activity is enhanced in postmortem PD brains [9], inducing ER-mediated RNF146 expression via small compounds might prevent pathological processes in PD. Importantly, dopaminergic neurons in the midbrain express ERs whose activation plays role in functional maintenance of dopaminergic neurons [16]. In addition to RNF146 expression, ER activation is involved in diverse biological processes [17]. For instance, ER agonists can prevent neurodegenerative processes through modulating cell survival mechanisms, synaptic organization, regenerative responses, and neurogenesis [17]. Inhibitory action of rhododendrin on brain microglia activation could suppress neuroinflammation and subsequent neurodegeneration in diverse brain disorders [17]. However, it is still unclear whether rhododendrin-mediated ER activation also modulates microglia and neuroinflammatory processes in addition to its direct protective role in dopaminergic neurons. 

At this point, it remains unknown how rhododendrin leads to ERβ activation. It could be through direct binding of rhododendrin to the ERβ receptor or through interacting with still unknown molecular targets that affect ERβ-related pathways. Although we showed that rhododendron-induced RNF146 expression is mediated by ERβ activation in vitro, the functional role of ERβ in RNF146 expression by rhododendrin treatment in vivo still requires further investigation. Biochemical and neuropathological characterization using ERβ knockout mice would be instructive in elucidating rhododendrin-ERβ-RNF146 pathway in vivo. Nevertheless, our results suggest that rhododendrin could be applied to enhance RNF146 expression possibly via ERβ activation in mouse brains. However, caution must be taken when applying ER agonists to treat neurodegenerative diseases because ER activation is associated with a potential risk for tumor growth [18,19]. Although there is no reported literature regarding adverse tumorigenic effect of rhododendrin, more thorough studies are required to obtain safety profiles of long-term rhododendrin treatment in vivo.

## 4. Materials and Methods

### 4.1. Chemicals and Antibodies

Rhododendrin was provided by the National Development Institute of Korean Medicine (NIKOM, Gyeongsan, Korea). Rhododendrin was purified from *Rhododendron brachycarpum*, and validated via high-performance liquid chromatography by NIKOM. 6-OHDA was purchased from Sigma. Tamoxifen was purchased from Selleck Chemicals (Houston, TX, USA).

The following primary antibodies were used: mouse antibody to RNF146 (N201/35, 1:5000, NeuroMab, Davis, CA, USA), mouse antibody to PAR (cat# 4335-MC-100, 1:3000, Trevigen, Gaithersburg, MD, USA), rabbit antibody to ERβ (cat#PA1-311, 1:3000, Invitrogen), rabbit antibody to tyrosine hydroxylase (NB300-109, 1:2000, Novus Biologicals, Centennial, CO, USA). For secondary antibodies, we used horse radish peroxidase (HRP)-conjugated sheep antibody to mouse IgG (cat# RPN4301, 1:5000, GE Healthcare, Pittsburgh, PA, USA), HRP-conjugated donkey antibody to rabbit IgG (cat# RPN4101, 1:5000, GE Healthcare), biotin-conjugated goat antibody to rabbit IgG (cat# BA-1000, 1:1000, Vector Laboratories, Burlingame, CA, USA), and HRP-conjugated mouse antibody to β-actin (cat# A3854, 1:10000, Sigma-Aldrich, St. Louis, MO, USA).

### 4.2. Purification of Rhododendrin from Rhododendron Brachycarpum

The dried leaves of *R. brachycarpum* (6 kg) were extracted with 10 L of MeOH for 3 h. The extract was filtered and concentrated using rotary vacuum drier to give MeOH extract (1265 g). The MeOH extract was suspended with distilled water (3000 mL) and partitioned with hexane, ethylacetate and butanol. The butanol soluble fraction (168.9 g) was subjected to chromatography on a diaion HP-20 resin column and eluted with a step gradient of H_2_O and MeOH (10:0 to 0:10, *v*/*v*), to give five fractions (RBE 1–5). RBE 3 was separated by silica gel (230-400 mesh, Merck, Germany) column eluted with a step gradient of chloroform:MeOH (10:1 to 2:1, *v*/*v*) to give eleven fractions (RBE 3-1–RBE 3–11). RBE 3-5 was purified by Sephadex LH-20 column eluted with 40% MeOH (*v*/*v*) to yield rhododendrin (102.9 mg). The structural identification of rhododendrin was based on ^1^H and ^13^C NMR spectroscopic data (JEOL-ECX 500, JEOL Ltd., Tokyo, Japan) and purity (99.8%) was analyzed with an Agilent 1260 HPLC system (Agilent Inc., Santa Clara, CA, USA).

Rhododendrin: White amorphous power; EI-MS *m*/*z* = 328 [M]+, molecular formula C_16_H_24_O_7_; ^1^H-NMR (500 MHz, CD_3_OD) *δ* 7.02 (2H, d, *J* = 8.6 Hz, H-2′, 6′), 6.66 (2H, d, *J* = 8.3 Hz, H-3′, 5′), 4.31 (1H, d, *J* = 7.7 Hz, Glc-1), 3.90-3.84 (2H, m, H-2, Glc-6a), 3.68 (1H, dd, *J* = 11.7, 5.4 Hz, Glc-6b), 3.36-3.29 (2H, m, Glc-3, Glc-4), 3.24 (1H, m, Glc-5), 3.16 (1H, t, *J* = 8.0 Hz, Glc-2), 2.62-2.56 (2H, m, H-4), 1.84 (1H, m, H-3a), 1.68 (1H, m, H-3b), 1.18 (3H, d, *J* = 6.3 Hz, H-1); ^13^C-NMR (125 MHz, CD_3_OD) *δ* 156.3 (C-4′), 134.8 (C-1′), 130.5 (C-2′, 6′), 116.1 (C-3′, 5′), 102.3 (Glc-1), 78.3 (Glc-3), 77.9 (Glc-5), 75.2 (C-2), 75.2 (Glc-2), 71.8 (Glc-4), 62.9 (Glc-6), 40.7 (C-3), 31.9 (C-4), 20.0 (C-1).

### 4.3. Cell Culture and Transfection

Human neuroblastoma SH-SY5Y cells (ATCC, Manassas, VA) were grown in DMEM containing 10% FBS (vol/vol) and antibiotics (penicillin-streptomycin 100 U/mL, ThermoFisher Scientific, Waltham, MA, USA). Cells were propagated in a humidified atmosphere consisting of 5% CO_2_/95% air and maintained at 37 °C. For transient transfections of the indicated vectors, X-tremeGENE HP transfection reagents (Roche, Mannheim, Germany) were used according to the manufacturer’s instructions. 

### 4.4. Plasmids

The RNF146 promoter luciferase reporter construct (pGL3-RNF146-Luc) was generated by subcloning the PCR-amplified RNF146 promoter (−1941 bp~−1 bp from the transcription start site; amplified from genomic DNA extracted from SH-SY5Y cells) into the pGL3 luciferase backbone (Promega). The CRISPR-cas9 construct targeting human ERβ was generated by cloning the sgRNA sequence for ERβ (5′-CACCGTCTGCAGCGATTACGCATC-3′) into a lentiCRISPR-v2 plasmid (Addgene plasmid #52961). Construct integrity was validated by sequencing. pLKO-shRNA targeting RNF146 and pLKO-shRNA targeting dsRed constructs [8] were previously described. 

### 4.5. Western Blotting

Total protein lysates were prepared by adding lysis buffer (1% Nonidet P40 in phosphate-buffered saline (PBS), pH 7.4) supplemented with protease/phosphatase inhibitors to SH-SY5Y cells that had been washed briefly with ice-cold PBS. After three freeze-thaw cycles in dry ice, samples were centrifuged at 14,000× *g* for 30 min. Next, the supernatants were mixed with 2× Laemmli buffer (Bio-Rad, Hercules, CA, USA) supplemented with β-mercaptoethanol (Sigma-Aldrich, St. Louis, MO, USA). After boiling the samples for 5 min, proteins were separated by SDS-PAGE and transferred to nitrocellulose membranes for immunoblotting. The blotted nitrocellulose membranes were stained with Ponceau (Sigma) to verify uniform protein transfer. Immunoblotting was performed with the designated antibodies and immunoreactive bands were visualized via chemiluminescence (Pierce). Densitometric analyses of the bands were performed using ImageJ (NIH, http://rsb.info.nih.gov/ij/).

### 4.6. Luciferase Assay

SH-SY5Y cells were transiently co-transfected with pGL3-RNF146-Luc and pRL-TK (Promega, Madison, WI, USA). Cells were harvested at 37 h following treatments with each compound and lysates were assayed for firefly luciferase activity using the Dual Luciferase Reporter Assay System (Promega, Madison, WI, USA) with a microplate luminometer (Berthold Technologies, Bad Wilddbad, Germany) according to the manufacturer’s instructions. Firefly luciferase levels were normalized to those of the Renilla control. As a negative control, cells were treated with 0.1% DMSO. Luciferase values for each chemical treatment were normalized to that of the DMSO control.

### 4.7. Real-Time Quantitative PCR

Total RNA was extracted with QIAzol Lysis Reagent (cat# 79306, QIAGEN, Hilden, Germany), then treated with DNase I to eliminate trace DNA contamination. cDNA was synthesized from total RNA (1.5 ug) using a first-strand cDNA synthesis kit (iScript cDNA synthesis kit, Bio-Rad). The relative quantities of mRNA expression were analyzed using real-time PCR (QuantStudio 6 flex Real-Time PCR System, Applied Biosystems, Foster City, CA, USA). SYBR Green PCR master mix (Cat# 4309155, Applied Biosystems) was used according to the manufacturer’s instructions. The relative mRNA expression levels of target genes were calculated by the ΔΔ*Ct* method [20] using GAPDH as an internal loading control. The primer sequences for real-time gene amplification are as follows:
hGAPDH: F-AAACCCATCACCATCTTCCAG, R-AGGGGCCATCCACAGTCTTCT;hRNF146: F-ATTCCCGAGGATTTCCTTGACA, R-GCTCATCGTACTGCCACCA;mGAPDH: F-TGGCCTTCCGTGTTCCTAC, R-GAGTTGCTGTTGAAGTCGCA;mRNF146: F-AGTCCTGTTCCAATACTGCACC, R-GAAGCACCCTTTACACACAGAT.


### 4.8. Chromatin Immunoprecipitation

Chromatin immunoprecipitation was carried out according to manufacturer’s instructions (Millipore) with the following modifications. Briefly, SH-SY5Y cells (treated with DMSO or Rhododendrin) were fixed with 1% formaldehyde for 10 min at 37 °C. Glycerol quenched samples were lysed in 1 mL of SDS buffer containing protease inhibitors. The lysates were incubated for 10 min on ice and sonicated to shear DNA. The samples were then centrifuged at 10,000× *g* at 4 °C for 10 min and the supernatant was collected. Precleared samples were incubated with either anti-ERβ, anti-histone antibodies, or rabbit IgG (rIgG)-agarose beads followed by a number of washes. Elutes were subjected to reverse crosslinking and DNA was recovered by phenol-chloroform-ethanol purification. PCR was performed using template DNA and the following primers:

Putative ERβ binding motif (TGACCT) within the RNF146 promoter (F-CGAGTAGCTGGGATTACAGGC; R-ACACACTTAAAGAGGTTCTCTGTA), RNF146 promoter-control region (F-GCGCAAGCATCACTGAACTA; R-TGTTGCATTTTGGGATTTCA), β-actin region (F-AGAGCTACGAGCTGCCTGAC; R-AGCACTGTGTTGGCGTACAG).

### 4.9. Cell Viability Assay

SH-SY5Y cells were plated in 6-well plates at a density of 0.5 × 10^6^ cells per well. Following transient transfection with the indicated constructs, cells were grown in DMEM containing low serum (2.5% FBS), with or without chemicals at the indicated concentrations, for the indicated durations. Next, the cells were harvested by trypsinization, thereby yielding single cell suspensions. The cells were washed twice with PBS and then resuspended in serum-free DMEM. Resuspended cells were mixed with an equal volume of 0.4% trypan blue (*w*/*v*) and incubated for 2 min at room temperature. Live and dead cells were counted using a Countess II Automated Cell Counter (Life Technologies, Bothell, WA, USA). 

### 4.10. Animal Experiments

All animal experiments were approved by the Ethical Committee of Sungkyunkwan University and were conducted in accordance with all applicable international guidelines (Approval number for this animal experiment: SKKUIACUC2017-05-06-1). Male C57BL/6N mice (3 months old, total 54 mice used to evaluate rhododendrin treatment in vivo (Appendix A), body weight = 26.3 g ± 0.164) were obtained from Orient (Suwon, Korea). Animals were maintained on a 12-h dark/light cycle in air-controlled rooms. Mice were provided ad libitum access to food and water. All efforts were made to minimize animal suffering and to minimize the number of animals used. Rhododendrin was administered to mice intraperitoneally. Rhododendrin administration (10 mg/kg body weight, i.p.) began on day 0 and was continued for 7 days, followed by stereological assessment of dopaminergic neuron counts. Intrastriatal injection of 6-OHDA was performed on day 3. Mice brains were prepared for analysis as described below. 

### 4.11. LC-MS/MS Measurement of Brain Rhododendrin

Mouse brain was extracted 20 min after the second treatment of rhododendrin to obtain approximate peak concentration of the compound. To avoid residual blood contamination, mice were anesthetized with pentobarbital (50 mg/kg, intraperitoneal injection) and intracardially perfused with PBS. Extracted mouse brains were further washed three times in ice cold saline and homogenized in two volumes of PBS (2 mL PBS per g brain tissue). To determine the concentration level of rhododendrin in the mouse brain, an LC-MS/MS method was applied with minor modification of a literature bioanalytical method [21]. Briefly, 100 µL of brain homogenates were mixed with 200 μL methanol with internal standard, vortexed, and centrifuged at 16,100× *g* (rcf) for 15 min at 4 °C. Supernatant (2 µL) was directly injected into the LC-MS/MS system. In detail, the LC–MS/MS system consisted of an Agilent HPLC system (1290 Infinity, Agilent Technologies, Santa Clara, CA, USA) and Agilent 6490 QQQ mass spectrometer with a negative electrospray ionization (ESI^-^) Agilent Jet Stream ion source (Agilent Technologies, Santa Clara, CA, USA). To achieve a good separation of rhododendrin and salicin (IS) from the endogenous substances in the brain, a Luna amino column (150 mm × 2.0 mm, 3 μm, Phenomenex, Torrance, CA, USA) was used using the mobile phase of water and acetonitrile (20:80, *v*/*v*) at a flow rate of 0.2 mL/min. Multiple reaction monitoring (MRM) in the ESI- mode was selected as follows: Rhododendrin, 327.2→164.7; IS (salicin), 285.1→123.0. The data was acquired using the Mass Hunter software (version A.02.00; Agilent Technology, Santa Clara, CA, USA). The calibration curve for rhododendrin was linear over the range from 0.02 to 50 μM (correlation coefficient *r* > 0.996). Finally, the measured concentration in the brain was expressed as a unit of rhododendrin amount (μg)/g brain (please see the Appendix A).

### 4.12. Intrastriatal Injection of 6-OHDA

For stereotaxic injection of 6-hydroxy dopamine (6-OHDA, 8 µg), three-month-old C57/BL6N mice treated with either rhododendrin or DMSO for four days were anesthetized with pentobarbital (60 mg/kg). The 6-OHDA injection procedure was performed as described previously [22], but with some modifications. Briefly, an injection cannula (26.5 gauge) was inserted stereotaxically into the striatum (anteroposterior, 0.5 mm from bregma; mediolateral, 2.0 mm; dorsoventral, 3.0 mm) and unilaterally inserted into the right hemisphere. Drug infusion was performed at a rate of 0.2 µL/min. A total of 2 µL of 6-OHDA (4 µg/µL in sterile PBS) was injected into each mouse. After the final injection, the injection cannula was maintained in the striatum for an additional 5 min to ensure complete absorption of the chemical. The cannula was then slowly removed from the mouse brain. The head skin was closed by suturing. Wound healing and recovery were monitored following the surgery. For stereological analysis, animals were perfused intracardially with ice-cold PBS 4 days after intrastriatal 6-OHDA injection. Next, tissue was fixed with 4% paraformaldehyde. Mouse brains were removed and processed for immunohistochemistry.

### 4.13. Preparation of Tissues for Immunoblotting

Mice were euthanized by cervical dislocation. Mouse brain subregions (ventral midbrain, VM) were located following procedures described previously [23]. Mouse brain tissues were homogenized in lysis buffer [10 mM Tris–HCl, pH 7.4, 150 mM NaCl, 5 mM EDTA, 0.5% Nonidet P-40, 10 mM Na-β-glycerophosphate, Phosphate Inhibitor Cocktails I and II (Sigma), and a complete protease inhibitor mixture (Roche)] using a Diax 900 homogenizer. Five milliliters of lysis buffer per gram of brain tissue was used for homogenization. After homogenization, samples were rotated at 4 °C for 30 min to ensure complete lysis. The homogenates were then centrifuged at 52,000× *g* (rcf) for 20 min, and the resulting supernatants were collected. Protein levels were quantified using the BCA Protein Assay Kit (Pierce) with BSA standards. Proteins were then subjected to immunoblotting with the antibodies of interest. Immunoreactive bands were visualized with an enhanced chemiluminescence kit (Pierce). Densitometric analyses of protein bands were performed using ImageJ (NIH, http://rsb.info.nih.gov/ij/).

### 4.14. TH Stereological Cell Counting

After the scheduled treatments with rhododendrin (10 mg/kg body weight, daily intraperitoneal administration), animals that received intrastriatal injection with 6-OHDA (intoxication model) or PBS (controls) were anesthetized with pentobarbital (50 mg/kg, intraperitoneal injection) and perfused with PBS. Next, the tissue was fixed with 4% paraformaldehyde (*w*/*v* in PBS). Brains were post-fixed overnight with 4% paraformaldehyde and subsequently cryoprotected overnight in 30% sucrose in PBS (*w*/*v*). Coronal sections (thickness of 40 µm) were cut through the brain including the substantia nigra. Every fourth section was used for analysis. For analysis of tyrosine hydroxylase (TH) expression, sections were incubated with a 1:1000 dilution of rabbit polyclonal anti-TH (Novus) antibody followed by sequential incubations with biotinylated goat anti-rabbit IgG and streptavidin-conjugated horseradish peroxidase (HRP) using a Vectastain ABC kit (Vector Laboratories, Burlingame, CA, USA) according to the manufacturer’s instructions. To visualize TH-positive cells, 3,3-diaminobenzidine (DAB, cat# D4293, Sigma) was used as an HRP substrate. Immunostained brain sections were counterstained with Nissl. The total number of TH-positive neurons in the substantia nigra pars compacta was determined using the Optical Fractionator probe in Stereo Investigator software (MicroBrightfield, Williston, VT, USA). All stereological counting was performed in a manner blinded to mouse treatments.

### 4.15. Statistics

Quantitative data are presented as mean ± SEM. Power analysis was performed using G*Power 3.1 software (Franz Faul, Kiel, Germany) to determine approximate sample sizes for tyrosine hydroxylase stereological counting. On the basis of mean difference from our preliminary experiments, a total sample size of four mice was calculated to potentially obtain a significant difference (effect size *f* = 22.42 for 45% mean difference; α = 0.05). Statistical significance was assessed either via an unpaired two-tailed Student’s t-test (two-group comparisons) or an ANOVA test with Tukey’s HSD post-hoc analysis (comparisons of more than three groups). Differences with a *p* value < 0.05 were considered significant. GraphPad Prism software (GraphPad Software, Inc., San Diego, CA, USA) was used for preparation of all plots and all statistical analyses.

## Figures and Tables

**Figure 1 ijms-20-01772-f001:**
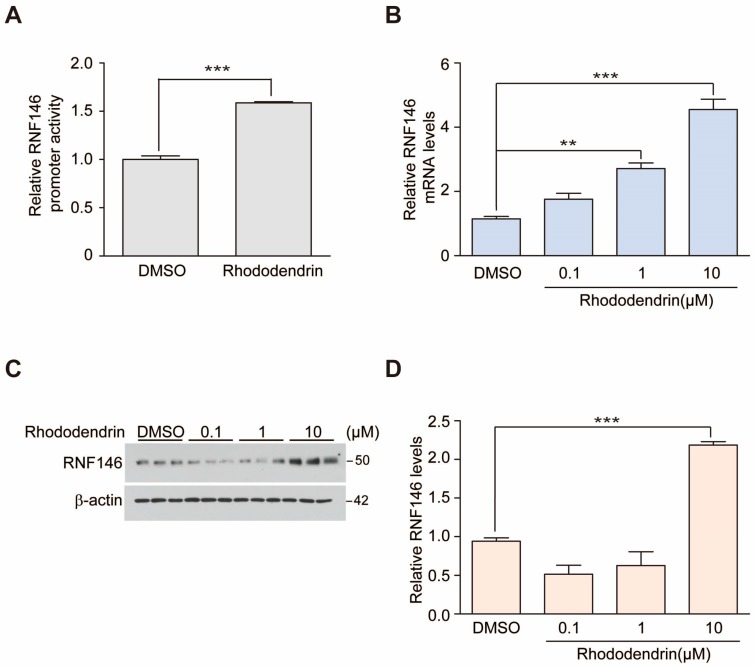
Rhododendrin increases RNF146 expression. (**A**) Quantification of relative RNF146 promoter activity in SH-SY5Y cells transfected with pGL3-RNF146-Luc and pRL-TK for 24 h, followed by treatment with 10 µM of rhododendrin for 37 h (*n* = 6 per group). (**B**) Quantification of relative RNF146 mRNA levels normalized to GAPDH in SH-SY5Y cells treated for 37 h with 0.1, 1, and 10 µM of rhododendrin (*n* = 3 per group). (**C**) Representative Western blot showing RNF146 expression in SH-SY5Y cells treated for 37 h with the indicated concentrations of rhododendrin (0.1, 1, 10 µM). β-actin served as a loading control. (**D**) Quantification of relative RNF146 protein levels normalized to that of β-actin in SH-SY5Y cells treated for 37 h with 0.1, 1, and 10 µM of the rhododendrin (*n* = 3 per group). Data are expressed as mean ± SEM. ** *p* < 0.01 and *** *p* < 0.001, ANOVA test followed by Tukey’s post-hoc analysis.

**Figure 2 ijms-20-01772-f002:**
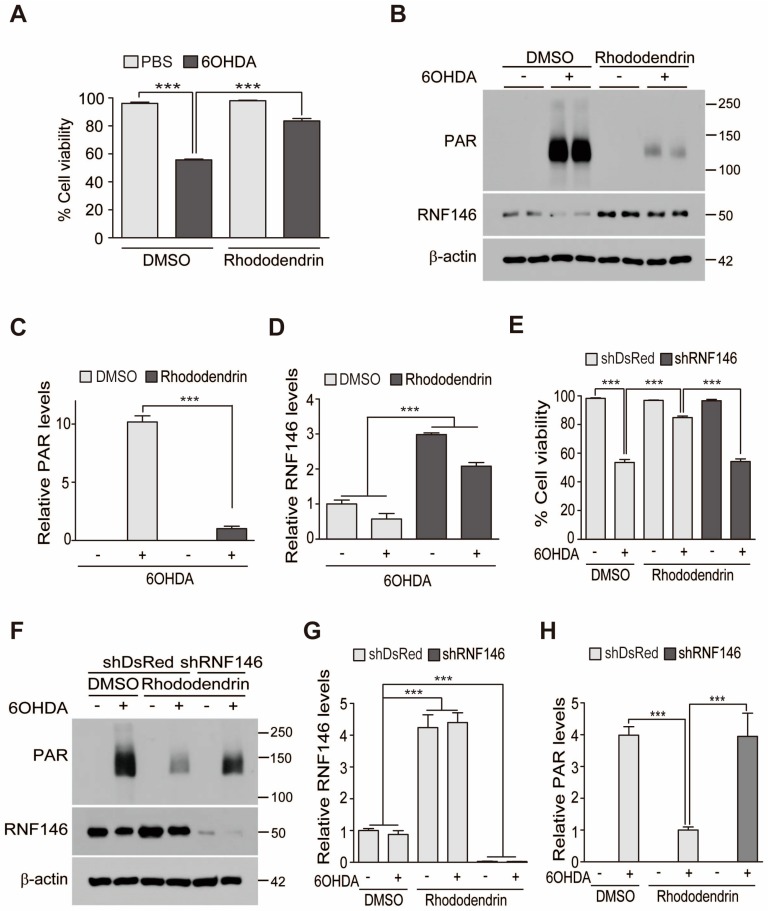
Rhododendrin-induced RNF146 expression protects SH-SY5Y cells from 6-OHDA toxicity. (**A**) Trypan blue exclusion cell viability assay demonstrating that rhododendrin pretreatment (10 µM, 37 h) increase cell survival after stimulation with 6-OHDA (70 µM, 16 h) (*n* = 6). (**B**) Representative immunoblots showing poly (ADP-ribose) (PAR), and RNF146 expression levels in SH-SY5Y cells challenged with 6-OHDA (70 µM, 30 min) following pretreatment with rhododendrin (10 µM, 37 h) or DMSO vehicle as a control. Quantification of relative PAR (**C**) and RNF146 (**D**) expression levels in the experimental groups in panel B. Values are normalized to those of β-actin (*n* = 3). (**E**) Trypan blue exclusion cell viability assay demonstrating that rhododendrin protection of SH-SY5Y cells from 6-OHDA (70 µM, 16 h) is RNF146-dependent. Cell survival caused by rhododendrin (10 µM, 37 h) was abolished by transient transfection of shRNA against RNF146 (61 h). shRNA against DsRed (shDsRed) was used as an shRNA transfection control (*n* = 6). (**F**) Representative immunoblots showing poly (ADP-ribose) (PAR), and RNF146 expression levels in SH-SY5Y cells transfected (61 h) with the indicated shRNA constructs and subsequently challenged with 6-OHDA (70 µM, 30 min) with or without pretreatment with rhododendrin (10 µM, 37 h) or DMSO vehicle as a control. Quantification of relative RNF146 (**G**) and PAR-conjugated protein (**H**) expression levels in the experimental groups in panel **F**. Values are normalized to those of β-actin (*n* = 3). Quantified data are expressed as mean ± SEM. Statistical significance was determined by ANOVA test with Tukey’s post-hoc analysis, *** *p* < 0.001.

**Figure 3 ijms-20-01772-f003:**
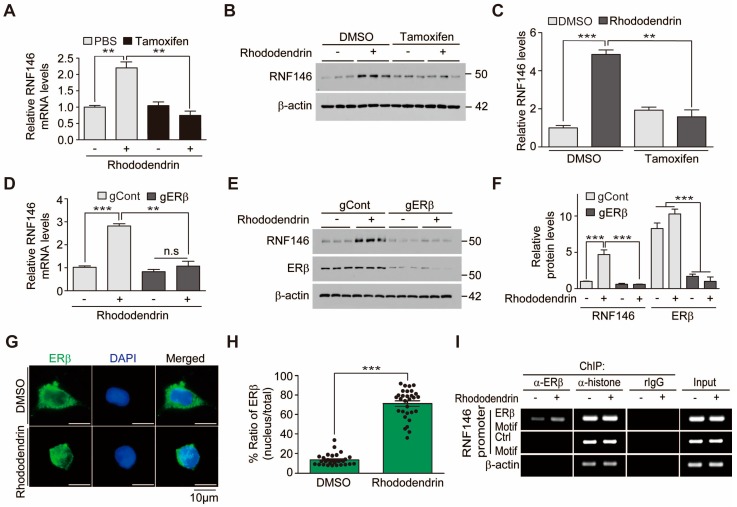
RNF146 expression induced by rhododendrin is mediated by ERβ activation. (**A**) Quantification of relative RNF146 mRNA expression levels (normalized to those of GAPDH) in SH-SY5Y cells treated with rhododendrin (10 µM, 37 h) in the presence or absence of the ER inhibitor tamoxifen (1 µM, 8 h) determined by RT-qPCR (*n* = 3). (**B**) Representative Western blot showing that rhododendrin (10 µM, 37 h)-mediated induction of RNF146 expression is blocked by tamoxifen treatment (1 µM, 8 h). (**C**) Quantification of relative RNF146 expression levels (normalized to those of β-actin) in the experimental groups in panel B (*n* = 3). (**D**) Quantification of relative RNF146 expression levels (normalized to those of GAPDH) in SH-SY5Y cells treated with rhododendrin (10 µM, 37 h) determined by RT-qPCR. ERβ expression was knocked down by CRISPR-cas9-mediated deletion of ERβ (85 h). sgRNA to EGFP was used as CRISPR-cas9 transfection control (*n* = 3). (**E**) Representative Western blot showing that rhododendrin (10 µM, 37 h)-mediated induction of RNF146 expression is abolished by CRISPR-cas9 mediated deletion of ERβ (85 h). (**F**) Quantification of relative RNF146 and ERβ expression levels (normalized to those of β-actin) in the experimental groups in panel E (*n* = 3). (**G**) Representative immunofluorescence images showing nuclear translocation of ERβ in response to rhododendrin treatment (37 h) in SH-SY5Y cells. (**H**) Relative distribution of ERβ in the nucleus as normalized to ERβ in the total cell area (*n* = 30 cells from three independent experiments). (**I**) Chromatin anti-ERβ immunoprecipitation (ChIP) of a putative ER responsive element (ERβ motif) within the RNF146 promoter region determined by PCR using specific primers. Non-ER responsive elements within RNF146 promoter (Control motif) and the β-actin region were used as negative controls. Immunoprecipitation using either anti-histone antibodies or rabbit IgG was included as a ChIP experimental control. Note the increase of ERβ occupancy of the RNF146 promoter ERβ motif induced by rhododendrin treatment (10 µM, 37 h). Similar results were obtained from two independent experiments. Quantified data are expressed as mean ± SEM. Statistical significance was determined by unpaired two-tailed Student’s *t*-test (H) or ANOVA test with Tukey’s post-hoc analysis (**A**,**C**,**D**,**F**), ** *p* < 0.01, and *** *p* < 0.001.

**Figure 4 ijms-20-01772-f004:**
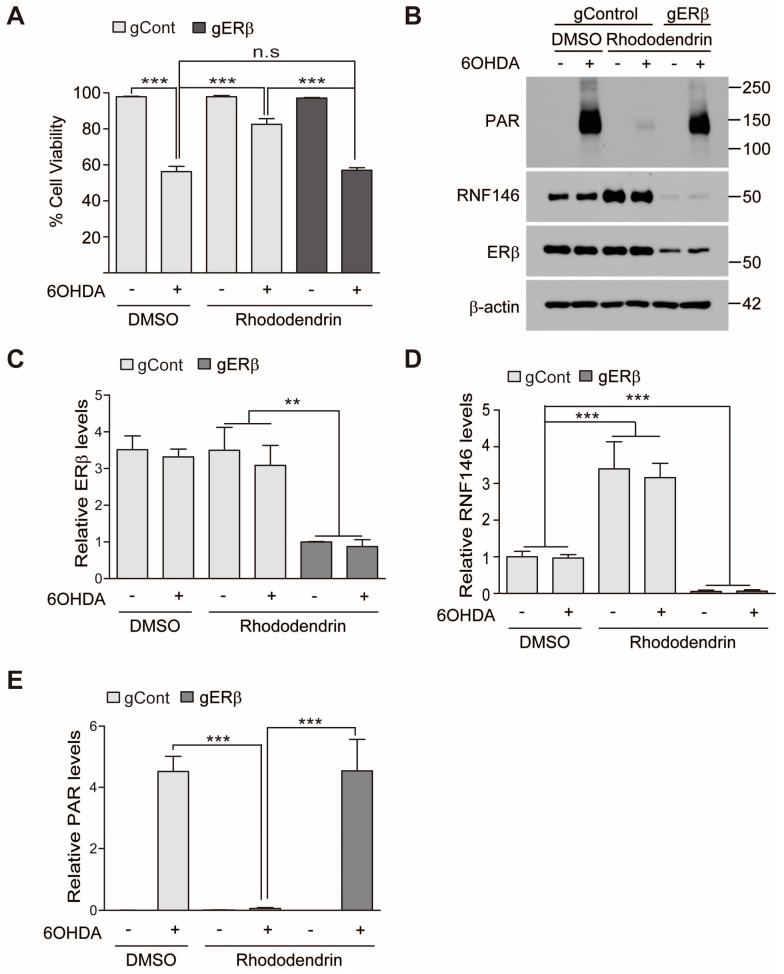
Rhododendrin prevents 6-OHDA cell toxicity via ERβ activation. (**A**) Assessment of cell viability in SH-SY5Y cells transfected with CRISPR-cas9 construct targeting ERβ (85 h) and subsequently treated with rhododendrin (10 µM, 37 h) or DMSO as control determined by a trypan blue exclusion assay. Oxidative stress induced cell death was induced by treatment with 6-OHDA (70 µM, 16 h) (*n* = 6). (**B**) Representative Western blot of PAR, RNF146, and ERβ in SH-SY5Y cells with the indicated transfections and treatments. SH-SY5Y cells were transfected with CRISPR-cas9 constructs (85 h) followed by 37 h treatment of DMSO or 10 µM rhododendrin for RNF146 induction. 6-OHDA (70 µM) challenge for PAR activation was given for 30 min. Quantification of relative ERβ (**C**), RNF146 (**D**), and PAR-conjugated proteins (**E**) expression levels in the experimental groups in panel B. Values are normalized to those of β-actin (*n* = 3). Quantified data are expressed as mean ± SEM. Statistical significance was determined by ANOVA test with Tukey’s post-hoc analysis, ** *p* < 0.01, and *** *p* < 0.001.

**Figure 5 ijms-20-01772-f005:**
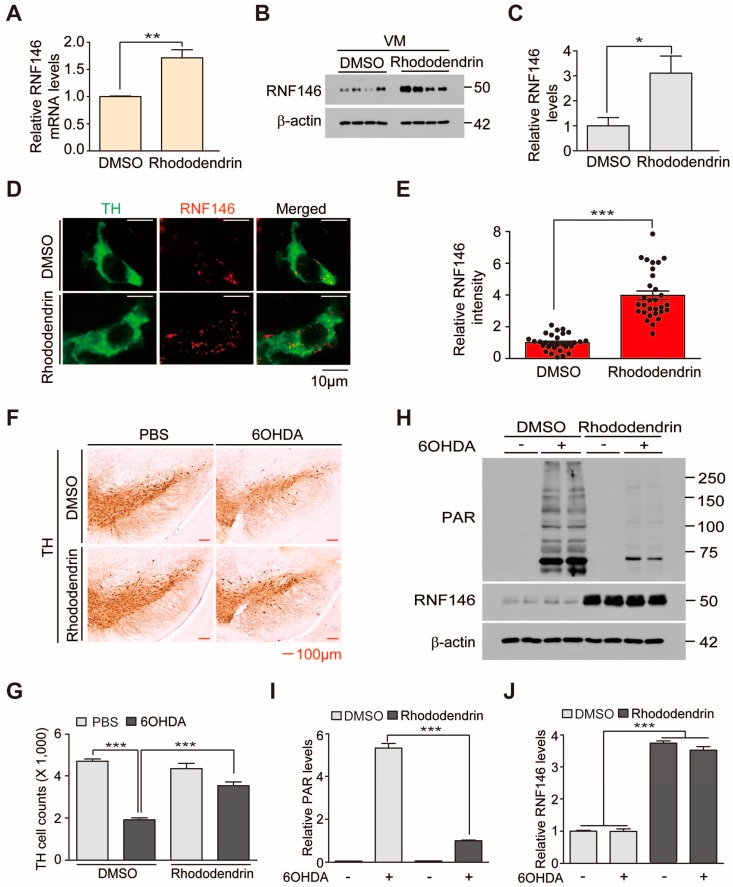
The RNF146 inducer rhododendrin prevents 6-OHDA induced dopaminergic cell death in vivo. (**A**) Quantification of relative RNF146 expression levels (normalized to those of GAPDH) in the ventral midbrain from mice treated with rhododendrin (i.p. 10 mg/kg/day, three days) as determined by RT-qPCR (*n* = 4). (**B**) Western blot analysis of RNF146 expression in the ventral midbrain from three-month-old mice treated with rhododendrin (i.p. 10 mg/kg/day, three days) or DMSO vehicle. VM, ventral midbrain. (**C**) Quantification of RNF146 protein expression levels in ventral midbrain from three-month-old mice treated with rhododendrin or DMSO for three days. Values are normalized to those of β-actin (*n* = four mice per group). (**D**) Representative confocal immunofluorescence images of TH (green) and RNF146 (red) expression using the indicated antibodies in ventral midbrain sections from three-month-old mice treated with rhododendrin or DMSO for three days. (**E**) Relative expression levels of RNF146 in TH-positive dopaminergic neurons in the rhododendrin-treated group as normalized to the DMSO group (*n* = 30 cells per each group from three mice). (**F**) Representative tyrosine hydroxylase (TH) immunohistochemical staining of the substantia nigra of 6-OHDA PD mice treated with rhododendrin (i.p. 10 mg/kg/day, seven days) or DMSO. 6-OHDA (8 µg, four days) was stereotaxically injected into the striatum (coordinates from bregma, L: −2.0, AP: 0.5, DV: −3.0 mm) to model dopaminergic neurodegeneration. (**G**) Stereological assessment of tyrosine hydroxylase (TH)-positive dopaminergic neurons in the substantia nigra pars compacta from the side of injection from the indicated mouse groups (*n* = four mice per group). (**H**) Western blot analysis of ventral midbrain (VM) protein expression of PAR and RNF146 in 3-month-old mice treated with rhododendrin (i.p. 10 mg/kg/day, seven days) or DMSO vehicle followed by intrastriatal 6-OHDA injection (37 h). (**I**) Quantification of PAR and RNF146 protein expression levels in ventral midbrain tissues from the indicated experimental groups in panel H. Values are normalized to those of β-actin (*n* = four mice per group). Quantified data are expressed as mean ± SEM. Statistical significance was determined by unpaired two-tailed Student’s *t*-test (**A**,**C**,**E**), or ANOVA test with Tukey’s post-hoc analysis (**G**,**I**), * *p* < 0.05, ** *p* < 0.01, and *** *p* < 0.001.

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
