# Peer review of "Rhododendrin-Induced RNF146 Expression via Estrogen Receptor β Activation is Cytoprotective Against 6-OHDA-Induced Oxidative Stress"

_ijms, 2019, doi:10.3390/ijms20071772_

Round 1

Reviewer 1 Report

The work is well written and planned. The conclusions are supported by the results obtained.

Minor

1. Figures 3G and 5D, concerning immunofluorescence experiments, are of poor quality.

2. The authors identify the optimal dose of rhododendrin to be used for treatments (10μM). On mice they directly use a dose of 10mg/kg ip. The authors should explain why the choice of this concentration. Also, I wonder if they checked rodhodendrin concentration on brain tissue. This could be a very important data.

Author Response

Pleae see the attached response to reviewers.

Reviewer 2 Report

Kim et al., reported that rhododendrin is able to induce RNF146 expression thereby preventing PARP1-dependent neurodegeneration in an animal model of Parkinson's disease (PD). They found that the increase of RNF146, after rhododendrin application, is due to estrogen receptor beta nuclear translocation and binding to the RNF146 promoter. Moreover, dopaminergic neurons lost was preventend in an animal model of PD after treatment with rhododendrin. The paper is well written and the results clearly presented.

Prior publication, I would like to suggest minor comments:

1) Figure 1: 1 μM of rhododendrin increases RFN146 mRNA level expression in SH-SY5Y but not  protein levels. Please try to check later time point. Please try also to check an intermediate concentration (5 μM).

2) Figure 2: please provide quantification of the WB showed in panel F.

3) Figure 4: please provide quantification of the WB showed in panel B.

Author Response

Please refer to the attached response to reviewers.

Reviewer 3 Report

The paper deals with an engaging study, with robust methodology, on the neuroprotective mechanism of rhododendrin thorough regulation of PARP1 overactivation in in vitro and in vivo PD models. However, the interconnection between the models and presentation of results is somewhat confusing, and the title and discussion should be improved. Addressing these issues is required before considering the publication of this article.

Major comments:

1.       The experimental design, in which the mechanistic in vitro study on the neuroprotective activity of rhododendrin are undertaken, without any confirmation from the mice experiment that this compound is distributed to the brain, is questionable. In addition, there is no published data on this matter and this has been expressed in the statement in lines 198-199 “It will be imperative to further investigate blood brain barrier permeability of rhododendrin and its potential application to several neurodegenerative disease animal models.” Based on the above, suggesting that rhododendrin caused any effects in the brain is not justified, please refer to sentences such as “Rhododendrin induced expression of RNF146 in cell culture and in vivo in mouse brains”. With this in mind, it is recommended to rephrase the relevant sentences, in the whole manuscript, in order to indicate that treatment with rhododendrin (not rhododendrin) caused effects in the brain.

Moreover, it is an inconsistency between the title, suggesting that rhododendrin-induced RNF146 expression via ET activation has been studied in a mouse model of PD and the presented results of the in vivo experiment. It would be valuable to investigate this mechanism for example in BERKO mice or OVX female animals. In addition, the title does not refer to the in vitro experiments, which results are mainly presented herein.

2.       A more detailed description of the animal experiment regarding the experimental design, including treatments and number of animals and their weight as well as the ID of the permit obtained from the Ethical Committee of Sungkyunkwan is required.

3.       In section 4.12. Preparation of tissues for immunoblotting, please provide the used ratio of tissue and buffer for preparing brain tissue homogenate as well as the centrifuge speed express in relative centrifugal force (RCF).

4.       Why was the effect of rhododendrin on RNF146 promoter activity in SH‐SY5Y cells assessed only for one concentration (10 μM)?

5.       There are some uncertainties and inaccuracies, as outlined below:

·         Line 80, please clarify with which group the change of “an approximately five‐fold induction by 10‐μM rhododendrin treatment” has been recorded?

·         Please provide labeling of statistical significance for changes in mRNF146 expression levels between cells treated with rhododendrin in the absence and presence of tamoxifen in Figure 3A, relative to the comment “ER inhibition by tamoxifen blocked RNF146 transcription induced by rhododendrin (Figure 3A)” (lines 107-108). Similarly, for changes in RNF146 protein expression in Figures 3B and C, as you stated that “Consistent with these results, tamoxifen‐mediated ER inhibition abolished RNF146 protein induction by rhododendrin (Figure 3B, C)” (lines 109-110).

·         There is an inconsistency between the statement that “CRISPR‐cas9‐mediated knockdown of ERβ resulted in failure of rhododendrin‐induced prevention of 6‐OHDA induced cytotoxicity (Figure 4A)” and the Figure 4A showing an increase in the cell viability in SH‐SY5Y cells under conditions of ERβ knockdown and 6‐OHDA-induced oxidative stress followed by treatment with rhododendrin. In regard to this relevant change please provide labeling of statistical significance.

·         Line 137 the information that “Intraperitoneal administration of rhododendrin for 4 days” is inconsistent with the caption of Figure 5A (“mice treated with rhododendrin (i.p. 10 mg/kg/day, 3 days)”

·         Line 166-167 the description that “Interestingly, inhibition of PARP1 overactivity was sufficient to block PD pathogenesis [4,6]” please clarify in regard to which PD model.

·         The caption of Figure 1B does not include information about the presented results for 0.1 µM and 1µM concentrations of rhododendrin.

·         The caption of Figure 3A does not refer to the mRNA level of RNF146 expression.

6.       Why are immunostaining results of TH and RNF146 expression not presented for mice treated with 6‐OHDA (Figure 5D), since TH+ cells are present in the substantia nigra of these animals based on results presented in Figure 5F?

Author Response

(The authors gave the same response as above.)

Round 2

Reviewer 3 Report

Please provide information on used procedures to avoid the measurement of rhododendrin from the residual blood and therefore to confirm its distribution to the brain.

Author Response

Reviewer comment: Please provide information on used procedures to avoid the measurement of rhododendrin from the residual blood and therefore to confirm its distribution to the brain.

Response: Following discussion with the collaborator and expert in pharmacokinetic analysis, we revised the corresponding method section by adding detailed information of mouse brain tissue processing. For your reference, the added information is as follows:

“To avoid residual blood contamination, mice were anesthetized with pentobarbital (50 mg/kg, intraperitoneal injection) and intracardially perfused with PBS. Extracted mouse brains were further washed three times in ice cold saline and homogenized in two volumes of PBS (2 ml PBS per g brain tissue). To determine the concentration level of rhododendrin in the mouse brain, an LC-MS/MS method was applied with minor modification of a literature bioanalytical method [21]. Briefly, 100 ul of brain homogenates were mixed with 200 ul methanol with internal standard, vortexed, and centrifuged at 16,100 g (rcf) for 15 min at 4 °C. Supernatant (10 ul) was directly injected into the LC-MS/MS system.”